# CCTFv2: Modeling Cyber Competitions

**DOI:** 10.3390/e26050384

**Published:** 2024-04-30

**Authors:** Basheer Qolomany, Tristan J. Calay, Liaquat Hossain, Aos Mulahuwaish, Jacques Bou Abdo

**Affiliations:** 1Cyber Systems Department, University of Nebraska at Kearney, Kearney, NE 68849, USA; 2School of Information Technology, University of Cincinnati, Cincinnati, OH 45221, USA; 3Department of Computer Science and Information Systems, Saginaw Valley State University, University Center, MI 48710, USA; 4School of Computing, Montclair State University, Montclair, NJ 07043, USA

**Keywords:** cybersecurity, cyber competition, teams, machine learning

## Abstract

Cyber competitions are usually team activities, where team performance not only depends on the members’ abilities but also on team collaboration. This seems intuitive, especially given that team formation is a well-studied discipline in competitive sports and project management, but unfortunately, team performance and team formation strategies are rarely studied in the context of cybersecurity and cyber competitions. Since cyber competitions are becoming more prevalent and organized, this gap becomes an opportunity to formalize the study of team performance in the context of cyber competitions. This work follows a cross-validating two-approach methodology. The first is the computational modeling of cyber competitions using Agent-Based Modeling. Team members are modeled, in NetLogo, as collaborating agents competing over a network in a red team/blue team match. Members’ abilities, team interaction and network properties are parametrized (inputs), and the match score is reported as output. The second approach is grounded in the literature of team performance (not in the context of cyber competitions), where a theoretical framework is built in accordance with the literature. The results of the first approach are used to build a causal inference model using Structural Equation Modeling. Upon comparing the causal inference model to the theoretical model, they showed high resemblance, and this cross-validated both approaches. Two main findings are deduced: first, the body of literature studying teams remains valid and applicable in the context of cyber competitions. Second, coaches and researchers can test new team strategies computationally and achieve precise performance predictions. The targeted gap used methodology and findings which are novel to the study of cyber competitions.

## 1. Introduction

A team is trivially considered to be a group of individuals engaged in or associated with one specific activity, but the literature has shown that it incorporates many complex dimensions, which are not always obvious. Structural entropy [1] describes the structural diversity of correlation-based communities within a network, which gives richer insights than the traditional interconnectedness of a network’s connected components [2]. This means that networks, including teams, are more than a group of individual nodes, which can be characterized with low structural entropy production and maximum entropy production. This aligns with Aristotle’s “the whole is greater than the sum of its parts”, where many disciplines have branched out to study the different dimensions involved in the creation and performance of teams [3], such as organizational sociology [4], anthropology [5], organizational behavior [6], industrial psychology [7], sociometry [8], social network analysis [9] and others. Teams, in addition to machines, were the core of the first industrial revolution and are believed to be, in addition to data, the core of the next one [10,11,12]. Teamwork has a significant ability to enhance performance by enhancing quality (detecting and recovering errors) [13], increasing productivity (facilitating specialization through division of labor) and fostering innovation [14]. Thus, understanding and optimizing team performance is critical for the success of organizations, companies, large-scale projects and competitive teams.

The U.S. Joint Special Operations Task Force activated the concept Team Of Teams [15] to address the growing threat of Al Qaeda in Iraq back in 2004. They realized that despite its enormous resources, best fighters, intelligence, training and technology and expertise, the U.S. Joint Special Operations Task Force struggled to overcome against the unstructured operations of Al Qaeda, which functioned as a decentralized network of agile teams, allowing them to strike quickly (achieving maximum entropy production), to reconfigure immediately and to integrate their actions globally in the complex environment in Iraq which was far from what the military had organized and trained for. They found success after it shifted from a command-and-control system to a Team of Teams model, in which units collaborated the same way individual members do on small teams.

It became clear that teamwork can enhance performance, but it can also induce noise and create errors resulting in dysfunctionality [13] and wasted energy, thus resulting in the conclusion that there is no maximum entropy production principle [16]. Multiple dimensions influence team performance, such as power dynamics [17], leadership, culture, organizational processes [18], interpersonal communication [19], team cohesion [20] and team formation (member selection and role combination) [21].

The more experienced French national team of FIFA’s 2002 world cup failed to win any match after winning the 1998 world cup. Interestingly, more than 80% of the 1998 team [22] participated in 2002 [23], and the same player (Zinedine Zidane) captained both teams. The major difference between the two teams was formation, which is the number of defenders, midfielders and forwards. The French national team used the very defensive 4–3–3 (and sometimes extremely defensive 4–5–1) combination in 1998 versus the more well-rounded 4–2–3–1 in 2002 [24]. Experts refer the significant difference in performance to France’s surprising formation in the 1998 finals and their opponents’ underpreparation for such extreme formations [25].

Out of the 11 football (soccer) players, each team is configured differently as an expression of the coach’s strategy. Accordingly, the team is configured into a combination of roles, where each role requires a specific skillset. The division of teams into role configurations and assigning members into roles is common in professional team formations. Team formation strategies have received considerable attention in competitive sports [26,27,28,29] and project management [30,31] but very rarely in cybersecurity.

### 1.1. Approach and Novelty

Cybersecurity team formation should leverage the years of experience and knowledge accumulated by other fields for optimized performance; however, the literature fails to bridge this gap. Cyber team formation has not received sufficient investigation, and this research track is still in its infancy. This motivated us to investigate the impact of cyber team formation strategies on team performance in cyber competitions. We start the theoretical approach by building a theoretical model, discussed in Section 3.1, that is grounded in the literature of team formation and performance beyond it. This model helps identify which team performance theories are applicable to the context of cyber competitions.

The second approach is computational, where we computationally model a cyber competition (blue team/red team exercise). Each of the teams has two distinct roles, and team performance is measured based on the used configuration (the number of players per role). To eliminate biases due to players’ skills, we used Agent-Based Modeling as an approach for developing the roles. The measured performance is thus strictly due to team formation strategies and network properties. This approach extends CCTFv1 [32], which is discussed in Section 2.4.

The outcomes of both approaches are compared to cross-validate. This cross-validation two-approach methodology is novel in the study of cyber competitions. This methodology offers higher confidence in both the implementation and conceptualization of the computational model. It also offers higher confidence in the precision of the results as they align with the literature present in the theoretical approach. Consequently, the results are theoretically explainable compared to black-box computational models. Finally, this methodology allows coaches and researchers to test new team strategies computationally and achieve precise performance predictions.

### 1.2. Contribution and Relevance

Our contribution begins with the application of a theoretical model that is central to our study. This model provides a foundational framework that integrates both observed factors (such as team size and individual skills) and latent factors (like collective skills and social cohesion). Applying this theoretical model bridges the gap between theoretical constructs and practical computational analysis, enhancing our understanding of team dynamics in cybersecurity contexts.

Following this, we introduce CCTFv2, an advanced computational model that builds upon the theoretical framework to quantitatively analyze cyber team performance dynamics. This model is unique in its comprehensive consideration of various team formation strategies and their direct impact on team effectiveness in cyber competitions.

Additionally, we have employed Structural Equation Modeling (SEM) as a part of our machine learning approach. SEM is crucial for this study as it allows for the analysis of structural relationships between the measured variables and latent constructs. This method is particularly suitable because it estimates multiple and interrelated dependencies in a single analysis, providing a more holistic understanding of the data. SEM’s ability to differentiate between endogenous (dependent) and exogenous (independent) variables adds depth to our analysis, enabling us to dissect the intricate interplay of factors affecting team dynamics with greater precision and clarity.

Lastly, our findings offer empirical evidence demonstrating the critical role of strategized team formations in cybersecurity. This challenges the current norms in corporate cybersecurity and cyber competition teams, emphasizing the need for more strategic approaches to team formation. Thus, our research makes a substantial contribution to both the academic field and practical realms, offering actionable insights for organizations looking to enhance their cybersecurity efforts through effective team formation strategies.

The remainder of this paper is organized as follows: Section 2 covers some of the important works which this work builds on. In Section 3, the theoretical model is proposed and the computational model is designed. Section 4 illustrates and analyzes the results before Section 6 concludes this work.

## 2. Related Work

In this section, we will delve into relevant works pertaining to our study in team formation, cyber competitions, Structural Equation Modeling (SEM), network attack modeling and Agent-Based Modeling. Specifically, our research focuses on the intersection of team formation and cyber competitions, aiming to address the unique challenges and opportunities presented in this domain. Cyber competitions, such as Capture the Flag (CTF) events, provide a platform for individuals to enhance their cyber skills and cybersecurity awareness, making it crucial to understand how team formation strategies impact performance in these contexts. Through our investigation, we seek to uncover effective strategies for optimizing team composition and dynamics to excel in cyber competitions.

### 2.1. Team Formation

In [33], the author explores a method for investigating distributed cooperative cyber defense mechanisms against network infrastructure-oriented attacks such as network worms, botnets and Distributed Denial of Service. The approach combines discrete-event simulation, a multi-agent approach and packet-level simulation of network protocols to simulate cyber-attacks and protection mechanisms. The defense system is represented by teams of defense agents that can cooperate as components of different organizations and ISPs. The paper outlines the simulation environment’s common framework, implementation peculiarities and experiments conducted to investigate distributed network attacks and defense mechanisms. The results demonstrate the approach’s effectiveness and potential for improving cyber defense mechanisms.

In [34], the authors present a study to improve the effectiveness of Cybersecurity Operations Centers (CSOCs) by forming effective teams that can efficiently analyze threat signals. The current approach of forming teams based on the credentials and expertise of individual analysts results in low-performing teams. The proposed framework defines team requirements and selects individuals to form several collaborative teams that meet these requirements. The framework integrates optimization, simulation and scoring methods to form effective teams and introduces a new collaborative score metric to measure their effectiveness. Results from simulated experiments show the formation of effective teams whose collaborative scores are maximized and balanced. Additionally, the approach can identify high and low performers within the first few months of implementing the framework, thereby providing insights for team management.

In [35], the authors proposed a network infection model to simulate the spreading impact of various cyber-attacks within law enforcement networks linked through the National Data Exchange (N-DEx) system, the central informational hub located at the FBI. The model is designed to create a level of organization from the state level to the local level of law enforcement agencies, allowing for each organizational infection probability to be calculated and entered, making the model specific for determining the spread or outbreaks of cyber-attacks among law enforcement agencies at all levels. The study highlights the importance of verifying the completeness, timeliness, accuracy and relevancy of N-DEx information through coordination with the record-owning agency to satisfy the Advanced Permission Requirements. The proposed model can help detect weak points within an information structure when multiple topologies merge, allowing for more secure operations among law enforcement networks.

In [36], the authors describe the successful implementation of general architecture and conceptual models in the MCWS simulator, focused on a collaborative ASW mission conducted through autonomous underwater vehicles. The results obtained from the simulation experiments are interesting and demonstrate the potential of this approach for scalable solutions to complex scenarios, with the possibility of interoperability with other models. The MCWS simulator allows for experimental and sensitivity analysis, enabling the evaluation of influential parameters, high-order effects and quantifying uncertainties and experimental errors. The simulation experiments also allow for the testing and evaluation of preventive action efficiency, mitigation procedures and reactions regarding their impact on operational scenarios, confirming the benefits of simulation as an investigative aid for cyber warfare within operational frameworks.

In [37], the author discusses a simulation-based approach to investigating distributed cooperative cyber defense mechanisms against network attacks. The method combines discrete-event simulation, a multi-agent approach and packet-level simulation of network protocols. Various experiments were conducted to investigate the effectiveness of defense mechanisms under different scenarios, such as network topology and configuration, the structure and configuration of attack and defense teams and the cooperation of defense teams. The experiments showed that team cooperation leads to a significant improvement in defense effectiveness.

In [38], the authors introduce OSIRIS (Organization Simulation In Response to Intrusion Strategies). This testbed can simulate cyber-attacks on organizations and analyze the potential damage based on organization size, cybersecurity expertise level and the proportion of communication. The objective is to provide a cost-effective and easy-to-repeat platform that allows clients to conduct various cyber-attack simulations without actual human tests. The method involves using a computer program replicating real-world organizations and running simulations to observe the effects of different cyber-attacks. The results demonstrate that OSIRIS can effectively simulate cyber-attacks and predict potential damage and can be used to conduct various outside-the-box experiments without constraints.

In [39], the authors propose an approach to simulate the defense mechanisms of information resources on the Internet using an agent-oriented simulation environment developed based on the OMNeT++ INET Framework. The environment enables the imitation of distributed attacks and defense mechanisms aimed at violating the accessibility of information resources and defending against these attacks. The study investigates the efficiency of the cooperation mechanisms of defense teams depending on the topology and configuration of the network, the structure and configuration of attack and defense teams and the parameters of cooperation of defense teams. The proposed approach’s results show efficiency in simulating defense mechanisms and analyzing the security of designed networks. The experiments demonstrate that the cooperation of several defense teams and the combined application of different defense methods greatly increase defense efficiency.

### 2.2. Cyber Competition

In [40], the authors aimed to understand the human dynamics of cybersecurity and the key factors that make a cyber defense team more or less effective in responding to and mitigating cyber-attacks. To achieve this, the researchers participated in data collection at the Mid-Atlantic Collegiate Cyber Defense Competition (MACCDC), where they collected data from wearable social sensors and used a 16-point teamwork instrument called OAT to assess teamwork and leadership behaviors in cyber defense. The study found that effective leadership and functional specialization within a team were important predictors of success in cyber defense. At the same time, face-to-face interactions emerged as a strong negative predictor of success. The results suggested that future research should focus on more concrete aspects of team coordination and evaluation in cyber defense teams.

In [41], the authors aimed to identify the characteristics of cybersecurity competition participants and determine the effectiveness of these competitions in attracting individuals to pursue careers in cybersecurity. The authors conducted a large-scale survey of 588 participants in the Cybersecurity Awareness Week (CSAW) competition, using measures of personality, interests, culture, decision making and attachment styles. They performed subgroup analyses to compare self-proclaimed hackers and non-hackers, males and females and cybersecurity employees versus students. Regression analyses were used to identify the variables that influenced the extent to which cybersecurity competitions effectively convinced participants to pursue a future career in cybersecurity. The results showed that participants who displayed higher self-efficacy, rational decision making style and more investigative interests were more likely to declare an interest in a career in cybersecurity after the competition. These findings have important implications for cybersecurity competition organizers who seek to attract individuals to the field.

In [42], the authors aimed to use data-driven analytical approaches to identify factors that predict the effectiveness of cybersecurity teams in securing networks, maintaining services and mitigating adversarial events. The authors analyzed data collected at the national finals and four regional events of the Collegiate Cyber Defense Competition, focusing on the teams’ experience, access to simulation-based training and functional role composition to predict team performance on four scoring dimensions. Bayesian analysis revealed that experience strongly predicted service availability, scenario injects and red team defense. Simulation training was also associated with good performance along these dimensions. High-performing and experienced teams clustered with one another based on the functional role composition of team skills. These findings provide insights into the effectiveness of challenge-based learning events and the stages of team development in cybersecurity.

In [43], the authors present the Cyber Forces Interactions Terrain (Cyber-FIT), a simulation framework designed to computationally model the performance of cyber teams operating in contested environments. The accurate projection of cyber mission forces and the simulation of desired effects have remained challenging tasks. The performance of military cyber teams in contested cyber terrains has been a subject of limited understanding. To address these issues, Cyber-FIT offers a comprehensive and extensible model that defines performance measures for cyber teams. The framework adopts an object-oriented and modular approach, enabling the incorporation of new measures without requiring significant architectural changes. This flexibility ensures that Cyber-FIT can adapt to emerging concepts and technological advancements. By leveraging Cyber-FIT, researchers can gain valuable insights into the performance of cyber teams in contested environments, contributing to a better understanding of cyber readiness and aptitude.

### 2.3. Structural Equation Modeling (SEM)

In [44], the authors provide a comprehensive yet accessible introduction to Structural Equation Modeling (SEM). This work is dedicated to outlining the fundamental elements of the SEM approach, tailored for researchers and students with a basic grasp of inferential statistics. The authors aim to bridge the gap between SEM and standard statistical methods prevalent in the social and behavioral sciences, such as correlation, multiple regression and analysis of variance. The focus is on demystifying SEM, explaining its components and applications in a non-technical manner. This approach encourages a broader understanding and application of SEM in various research scenarios, emphasizing its effectiveness in testing hypotheses about the relationships between observed and latent variables.

In [45], the authors present Structural Equation Modeling (SEM) as a theory-driven analytical approach for evaluating a priori specified hypotheses about causal relations among measured and latent variables. Their work explores SEM not just as a statistical technique but as a comprehensive analytical process, encompassing model conceptualization, parameter identification and estimation, assessment of data–model fit and potential model re-specification. The authors emphasize how SEM is instrumental in assessing the fit between correlational data, obtained from both experimental and non-experimental research, and pre-specified competing causal theories. They also discuss the role of software packages like AMOS, EQS, lavaan, LISREL, Mx and Mplus in the computational aspects of SEM and reference contemporary texts for further understanding of the methodology. Additionally, they provide specific criteria for applied studies using SEM.

In [46], the authors explore Structural Equation Modeling (SEM) as a methodology that differentiates measurement errors from the true scores of attributes, enabling the direct modeling of latent variables such as attitudes, IQ, personality traits and socio-economic status. They discuss the significant advancements in SEM since 1970 and its extensive application in various disciplines. The work emphasizes that SEM is often viewed as an extension of factor analysis in psychometrics, incorporating multiple indicators for a latent variable. The authors also address specific models within SEM, focusing on the general mean and covariance structures. A key point made is the distinction between exploratory factor analysis (EFA) and confirmatory factor analysis (CFA), where in CFA, zero loadings are determined a priori based on subject matter knowledge. This predetermined structure of zero loadings ensures that the model is identifiable without the need for rotational constraints.

### 2.4. Network Attack Modeling

In [47], the author highlights the challenges of developing effective cyber defense mechanisms against dynamically evolving attacks on networks using conventional fixed algorithms. The authors suggest that artificial intelligence methods, which offer flexibility and learning capabilities, are necessary for developing effective cyber defense capabilities. The article reviews current artificial intelligence practices and techniques and emphasizes the importance of integrating them into cyber defense systems. The authors aim to explain the use of these methods in cyber defense through current examples and by analyzing the adaptation and role of technology and methodology in cyberspace defense. The article highlights the potential of artificial intelligence in solving many cyber defense problems and the need for increased intelligence of defense systems.

In [48], the authors present the SCART framework, which enables the simulation of complex cyber-attacks and system failures in real-time systems. The method incorporates the SCART layer into simulation environments, simulating random component or subsystem failures and simple and complex security attacks. The authors validated the effectiveness of SCART by training machine learning algorithms to control a drone’s flight control system, resulting in increased accuracy and lower false-positive rates. The experiments showed that SCART could generate realistic cyber-attacks and different anomaly detection methods could successfully identify attacked and normal flights using SCART. Overall, the study demonstrated the potential of creating high-quality data for cyber-attacks using SCART, which can help develop deep learning models for real-time anomaly detection with a low false-positive rate.

In [49], the authors discuss the potential of using multi-agent systems in cybersecurity and present the initial steps towards building a general-purpose multi-agent system for a cyber defense network of computers. The goal is to detect threats, react to mitigate them and restore the appropriate level of service. The paper also includes a preliminary experiment to measure the burden of implementing this distributed defense mechanism regarding CPU and memory overload. The results suggest that the multi-agent system can scale up effectively. However, system performance may be affected when the number of simultaneous agents running on the same computer is high.

In [50], the authors aim to develop an effective control system for computer network security in the context of cyber-attacks. The proposed technique is based on constructing a system state space and a stack of control decisions using a finite Markov chain to calculate the probability of the security system being in a certain state at each control step. The developed algorithm can predict the number of iterations needed to manage the security system during a cyber-attack, generate control decisions automatically and determine the parameters most susceptible to abnormal deviations. The experimental results using a generated dataset demonstrate the high efficiency of the developed technique.

### 2.5. Agent-Based Modeling

In [51], the authors delve into the importance of network structure in team formation within multi-agent systems. Unlike past studies that overlooked network complexities or assumed simplistic structures, recent insights reveal the nuanced nature of real-world networks. With multi-agent systems growing in complexity, grasping network dynamics is essential for nurturing efficient agent communities. The study introduces an agent-based computational model to investigate team formation across various network topologies, including rings and stars. Through theoretical analysis and empirical experiments, the authors highlight diversity support as a critical factor influencing effective team formation. They show that scale-free networks, with their short average path lengths and hub-like structures, offer higher organizational efficiency compared to other network types, emphasizing the role of network structure in optimizing team dynamics.

In [52], the authors explore the role of network structure in team formation within multi-agent systems. Departing from previous studies that neglected network intricacies or assumed simplistic models, recent insights illuminate the intricate nature of real-world networks. As multi-agent systems grow in complexity, understanding network dynamics becomes vital for fostering efficient agent communities. The study presents an agent-based computational model to probe team formation across diverse network topologies, including rings and stars. Through theoretical analysis and empirical trials, the authors underscore diversity support as a pivotal factor shaping effective team formation. They demonstrate that scale-free networks, characterized by short average path lengths and hub-like structures, offer superior organizational efficiency compared to other network types, underscoring the significance of network structure in optimizing team dynamics.

In [53], the authors delve into the pivotal stage of team formation in deploying multi-agent teams. They raise pertinent questions regarding whether to prioritize team diversity or individual member strength in scenarios where agents coordinate through continuous voting. Specifically, they investigate whether a team composed of diverse yet weaker agents can outperform a uniform team of stronger agents. To address these inquiries, the authors propose a novel model, offering several key contributions: (i) demonstrating that a diverse team can surpass a uniform team under specific conditions; (ii) presenting optimal voting rules for diverse teams; (iii) conducting synthetic experiments highlighting the combined impact of diversity and strength on team performance; (iv) showcasing the utility of their model in addressing one of the most challenging tasks in artificial intelligence: Computer Go.

In [33], the authors examine an approach to investigating distributed cooperative cyber defense mechanisms targeted at network infrastructure-oriented attacks, such as Distributed Denial of Service, network worms and botnets. The approach entails agent-based simulation of cyber-attacks and cyber protection mechanisms, leveraging discrete-event simulation, a multi-agent approach and packet-level simulation of network protocols. The study explores various methods of counteracting cyber-attacks by representing attack and defense components as agent teams within a software simulation environment. These defense agent teams can collaborate as components of different organizations and Internet service providers (ISPs). The paper outlines the common framework and implementation specifics of the simulation environment, along with experiments aimed at probing distributed network attacks and defense mechanisms.

### 2.6. Research Gap and Previous Versions of CCTF Framework

In a previous version of the CCTF framework [32], we introduced the Collaborative Cyber Team Formation (CCTFv1) Simulation Framework to address the scarcity of empirical research on team strategy in cybersecurity. Using Agent-Based Modeling, the authors scrutinize team creation dynamics and output, emphasizing the impact of structural dynamics on performance while meticulously controlling other variables. Their findings underscore the significance of strategic team formations, an often-neglected aspect in corporate cybersecurity and cyber competition teams. Moving forward, the authors aim to validate and refine this computational model with empirical data collected from competitions, conducting them as controlled experiments to ensure methodological integrity. This work lays the groundwork for the CCTF framework, integrating theoretical, empirical and computational approaches for a comprehensive understanding of cyber team performance dynamics.

Compared to the above-related works in team formation, cyber competitions, Structural Equation Modeling (SEM), network attack modeling and Agent-Based Modeling, our study stands out by addressing a critical gap in the literature. While previous research primarily focuses on specific aspects such as defense mechanisms against cyber-attacks or participant characteristics in cybersecurity competitions, our work takes a holistic approach. We contribute by applying a theoretical model to understand team dynamics in cybersecurity contexts, bridging the gap between theoretical constructs and practical computational analysis. Additionally, our advanced computational model, CCTFv2, offers a comprehensive analysis of team performance dynamics, considering various team formation strategies and their direct impact on team effectiveness in cyber competitions. Leveraging Structural Equation Modeling (SEM), we delve deeper into understanding the structural relationships between measured variables and latent constructs, providing a more holistic understanding of team dynamics. Our findings offer empirical evidence highlighting the critical role of strategized team formations in cybersecurity, challenging existing norms and providing actionable insights for organizations seeking to enhance their cybersecurity efforts through effective team formation strategies.

## 3. Implementation Methodology

This work aims to assess the influence of different team factors on performance in cyber competitions. The considered influencing factors are as follows:**Team Formation:** The distribution of the resources and team members among roles and positions [32].**Team Size:** Larger teams tend to yield greater collective performance [54], indicating a direct proportionality between team size and performance.**Skills:** Teams with higher skills, encompassing talent [55] and training [18], typically outperform those with lesser skills, suggesting a direct correlation between members’ skills and team performance.**Collaboration:** More collaborative teams generally perform better [56], implying a direct relationship between collaboration and performance. However, talent and collaboration may sometimes be inversely proportional [57].**Leadership:** Teams exhibiting superior leadership and trust usually perform better than those facing leadership crises [58].**Social Influence:** Performance can be swayed by social pressures within the team, like interpersonal dynamics [59], or external pressures, such as crowd influence [60]. This encompasses factors like motivation, stress and discrimination.

We start by formalizing a theoretical model, based on the literature, that describes the relationship between the above influencing factors and performance. This model is discussed in Section 3.1. Section 3.2 discusses a proposed computational model which simulates the dynamics of a cyber competition.

### 3.1. Theoretical Model

Different factors contribute to the performance of a team. However, the overall competition result not only depends on each team’s performance (*Team Dynamics*) but also on the environment in which the competition is conducted. Let us consider a simplified theoretical model, illustrated in Figure 1, which involves two teams competing against each other. The performance of each team (depicted as *Team Dynamics*) in addition to environment (depicted as *Network Topology*) determine the competition’s result (depicted as *Observable 1*). *Observable 1* is observed, while *Team Dynamics* is latent.

We hypothesize, based on [61,62,63], that *Team Dynamics* is a latent factor that is determined by the above influencing factors (*Team Formation, Team Size, Skills, Collaboration, Leadership and Social Influence*). The last two factors will not be considered in this study. The composite relationship between the observed factors (*Team Formation, Team Size, Skills and Collaboration*) and the latent factor *Team Dynamics* are hypothesized to be as follows:*Team Size*, *Team Formation* and *Individual Skills* influence *Collective Skills* [64,65,66,67], which is a latent factor.*Team Size, Team Formation* and *Collective Skills* influence *Social Cohesion* [68,69,70,71], which is a latent factor.*Collaboration*, *Collective Skills* and *Social Cohesion* influence *Team Dynamics* [72,73,74].

Accordingly, the hypothesized theoretical model is illustrated in Figure 2, where the rectangular shapes represent observed factors and the elliptic shapes represent latent factors.

### 3.2. Computational Model

In this study, a cyber competition is modeled as a defender team safeguarding a network against an attacker team. Following the models established in [43,75], the network constitutes interconnected nodes, with only the peripheral nodes (Internet-facing) being accessible to attackers initially. On successfully overtaking a peripheral node, attackers can target linked nodes. Figure 3 depicts such a network, where peripheral nodes in the yellow region are exposed to attack attempts (black arrows). Non-peripheral nodes in the green region are unreachable directly but become accessible (blue arrows) once a peripheral node is compromised (red arrow).

In our model, each attacker assumes one of two distinct roles: a *scout* or an *exploiter*. The scout scans accessible nodes to detect vulnerabilities, with the probability of successful detection—given a node is vulnerable—set by the user and defined as:$$P\left(detect\;|\;node\;is\;vulnerable\right)=P_{scout}$$

Once a scout identifies a vulnerability, exploiters are notified to attempt to exploit this vulnerability and control the compromised node. The probability of successful exploitation—given a node is vulnerable—is set by the user as:$$P\left(exploit\;|\;node\;is\;vulnerable\right)=P_{exploiter}$$

Systematically, the scout performs the two initial hacking steps as defined by EC-Council [76], as illustrated in Figure 4, which are as follows:**Hacking step 1**: Reconnaissance is the footprint or information gathering step. This step usually includes passive attempts.**Hacking step 2**: The scanning step is when the attackers start using different ways to gather target information; this usually includes active attempts.

In parallel, the exploiter is systematically characterized as the role executing steps 3 through 5, which are as follows:**Hacking step 3**: Gaining access is the step when the attackers try all means to gain unauthorized access.**Hacking step 4**: Maintaining access is when the attackers attempt to continuously exploit the system.**Hacking step 5**: Clearing track is when the attackers delete the infection signs, aiming to evade attribution and detection.

The attacker team consists of *s* scouts and N−s exploiters, where *N* denotes the total team size. To mitigate the influence of team size, as outlined in Section 3, this study maintains constant and equivalent team sizes for both attacker and defender teams.

Agent-Based Modeling (ABM) is employed to curtail the impact of *skills*, *collaboration* and *social influence* (detailed in Section 3.2.1). Identical agents mitigate *skills*, while broadcasting communication and anonymous interactions limit *collaboration* and *social influence*, respectively. An ad hoc team arrangement, with a user-determined size (*N*), helps manage the *leadership* factor. Success in a cyber-attack hinges on its objectives; this study outlines three unique objectives and corresponding metrics.

**Objective 1**: Overtake as many nodes as possible.**Objective 2**: Overtake the whole network.**Objective 3**: Overtake the central nodes representing the core services such as the database.

Accordingly, we identify three distinct metrics for measuring the performance, which are as follows:**Metric 1**: A portion of the network, overtaken by the attackers. This metric is a decimal number ranging between 0 and 1.**Metric 2**: Whether the whole network was overtaken by the attackers. This metric is a Boolean.**Metric 3**: Whether the central nodes were overtaken by the attackers. This metric is a Boolean.

In our model, each defender can assume one of two distinct roles: a *detector* or an *interceptor*. The detector scans the network for compromised (exploited or overtaken) or vulnerable nodes. The probability of successfully identifying an infected node—given that it is indeed infected—is set by the user and defined as:$$P\left(detect\;|\;node\;is\;infected\right)=P_{detector-exploited}$$
while the probability of successfully identifying a vulnerable node—given that it is indeed vulnerable—is set by the user and defined as:$$P\left(detect\;|\;node\;is\;infected\right)=P_{detector-vulnerable}$$

Upon the detection of an infected node, interceptors are alerted to defend the network. The interceptor can either flag the infected node as untrusted—thus isolating it [77,78,79]—or recover the node based on user-defined action. The interceptor requires time $\Delta_{interceptor}$, set by the user, to carry out this action. The detector is systematically aligned with the first three steps of incident handling, as defined by EC-Council [80], as illustrated in Figure 4, which are as follows:**Incident handling step 1**: To observe is a continuous security monitoring step.**Incident handling step 2**: The orient step includes evaluating the local cyber threat landscape.**Incident handling step 3**: The decide step includes deciding on an action plan.
Similarly, the exploiter is systematically defined as the role that performs step 4, which is as follows:**Incident handling step 4**: The act step includes the remediation and recovery actions.

The team of defenders is composed of *d* detectors and N−d interceptors. The same measures implemented to limit the impact of the factors defined in Section 3 on the attacker team’s performance are implemented to limit the impact on the defender team’s performance.

#### 3.2.1. Study Parameters

In the NetLogo simulation (screenshot shown in Figure 5), teams are composed of two types of agents. A detector agent scouts the network and relays information back to the team’s knowledge base, and then a worker agent consults the knowledge base in order to decide which actions it should take.

The probability that an agent successfully completes a task was multiplied by the skill value of the agent, which is a decimal value between 0 and 1. For each agent, their skill was a random value chosen from a normalized distribution.

The collaboration of a team was represented using a set of variables that corresponds to delays between actions. Defender detector agents have two variables: Action Delay and Report Delay. Action Delay governs how often a detector can check one router in the network. Report Delay is how long it takes the agent to send information about a vulnerable or compromised router to the defender team’s knowledge base. Defender interceptor agents have only the Action Delay variable, which determines how often they can send a fix for a vulnerable or compromised router. The simulation logic is illustrated in Figure 6.

Two different techniques were used to investigate the parameter space of the model. The first set of simulations (size: 1470 runs) explored the effects of changing a single parameter against a baseline configuration, in order to sample a wide range of possible networks. This dataset of runs is used to train the model discussed in Section 4 and detailed in Table 1. The second simulation (size: 48,600 runs) explored combinations of fine-tuned increments of the model parameters, to better examine how the systems interact. This dataset of runs is used to test and validate the model discussed in Section 4 and detailed in Table 2. Both datasets in addition to the NetLogo simulation are publicly accessible [81].

## 4. Results

The observed results are mapped over the theoretical model as a multi-layer structural equation model in Figure 7. Team Dynamics is calculated as a latent factor for each of the two teams, and both influence Observable 1. This validates the proposed theoretical model and cross-validates the computational model.

Structural Equation Modeling relies on several statistical tests to determine the adequacy of model fit to the data. The chi-square test indicates the amount of difference between expected and observed covariance matrices. A chi-square value close to zero indicates little difference between the expected and observed covariance matrices. If model fit is acceptable, the parameter estimates are examined. The Root Mean Square Error of Approximation (RMSEA) is related to residuals in the model. RMSEA values range from 0 to 1 with a smaller RMSEA value indicating better model fit. Acceptable model fit is indicated by an RMSEA value of 0.06 or less. The reported RMSEA value for our model is below 0.001, which is considered a good fit.

The generated structural equation model was trained using the dataset described in Section 3.2.1 and resulted in 62 “Degrees of freedom” and 0.000 as “*p*-value (Chi-square)”. The model was compared to the test dataset described in Section 3.2.1 and resulted in the same “Degrees of freedom” and “*p*-value (Chi-square)”.

Figure 8 shows a two-layer structural equation model hiding the latent factors and presenting each variable’s role in determining Observable 1. Team collaboration, represented as reporting delay, is the most impactful in determining Observable 1, and this holds true for both teams. By this, we have cross-validated the theoretical model, computational model and structural equation model.

## 5. Result Analysis and Discussion

SEM analysis was used to determine the relationships between the latent constructs and the observed variables presented in this model. An evaluation of how well the overall model fits with the data was determined by a number of fit indices. The chi-square value for the model is close to zero ($X^{2}$ = 0.0001), indicating a good fit to the data. However, it should be noted that chi-square statistics tend to depend on sample size; hence, other fit indices were also assessed in addition to this one statistic.

Results indicated that the RMSEA value was 0.06, below the suggested threshold of 0.08, thus evidencing good fit of the model to the data. Additionally, the comparative fit index (CFI) was 0.996, and the Tucker–Lewis index (TLI) was 0.992, both surpassing the level of acceptance, which is 0.90, further indicating a good fit of the model.

The results also revealed that the observed factor of team collaboration was highly determinant of the competition outcome (Observable 1) with the parameter estimate of 77.78 for defenders and 1.617 for attackers, suggesting a significant effect of team collaboration on the competition outcome. Overall, the results of the SEM analysis confirm the hypothesized inter-relations among latent constructs and observed variables in the model, suggesting that the model demonstrates a good fit and accounts for a considerable amount of variance in competition outcomes.

## 6. Conclusions

Cyber teams are playing more important roles in organizations, whether corporate or federal. Establishing the U.S. Cyber Command and the Cybersecurity and Infrastructure Security Agency (CISA) is an example of how cyber teams are formed into independent specialized structures operating independently or in coordination with traditional teams. Understanding the performance of cyber teams is becoming extremely important and timely; however, few have investigated this question.

This study employs a cross-validating two-approach methodology. The first involves the computational modeling of cyber matches using Agent-Based Modeling. NetLogo models team members as cooperating agents competing across a network in a red/blue team match. The second approach leverages the literature on teams in sports management and military science to build a theoretical model describing the unique features of cyber teams.

The theoretical model has been validated against the computational model using Structural Equation Modeling. The results validated the proposed theoretical model, giving insights into the dynamics of cyber teams. Two key conclusions were drawn. First, the body of literature on teams is still valid and appropriate in the setting of cyber contests. Second, coaches and researchers may use computational tools to evaluate new team strategies and make exact performance forecasts.

Moving forward, our next endeavor involves validating and potentially calibrating this computational model using empirical data collected from a competition. To ensure the integrity of the study, the competition will be organized as a controlled experiment, effectively minimizing the influence of other factors that could introduce biases. This line of research is still in its early stages and more empirical experiments are needed; however, these current results are promising as they provide a foundation for answering new research questions.

## Figures and Tables

**Figure 1 entropy-26-00384-f001:**
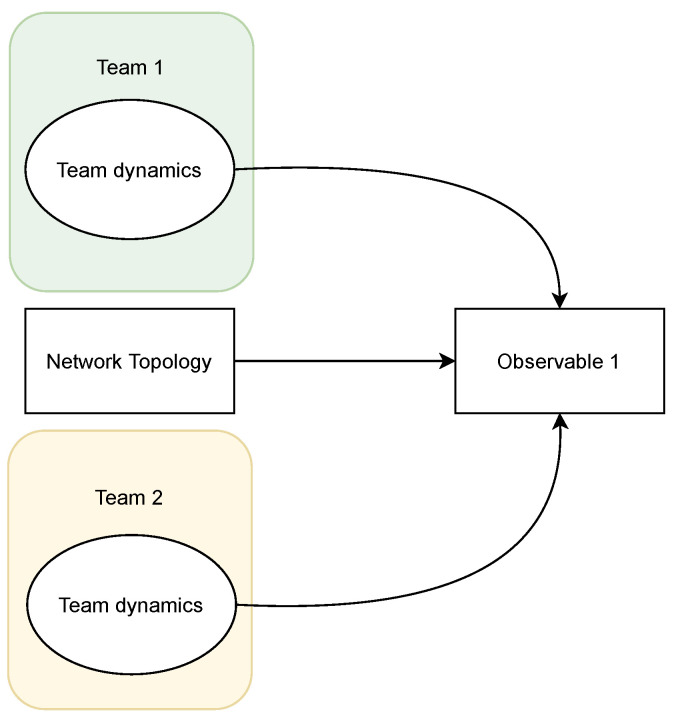
Simplified theoretical model. The match result (Observable 1) is simplistically formulated as the interaction between the two teams (Team Dynamics) and the environment in which the match is played (Network Topology).

**Figure 2 entropy-26-00384-f002:**
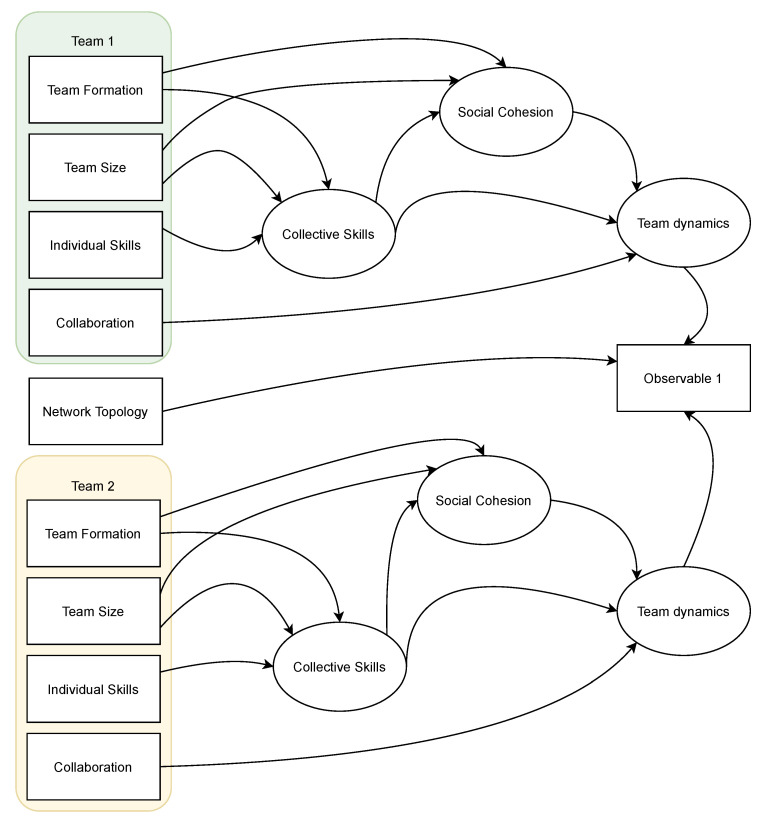
Proposed theoretical model. The match result (Observable 1) is formulated as the interaction between the two teams and the environment in which the match is played (Network Topology). The squares represent the SEM measured variables, where those to the left are controlled variables (inputs) and those on the right are outputs. The ovals represent the SEM latent variables. This model is constructed based on the literature of team performance.

**Figure 3 entropy-26-00384-f003:**
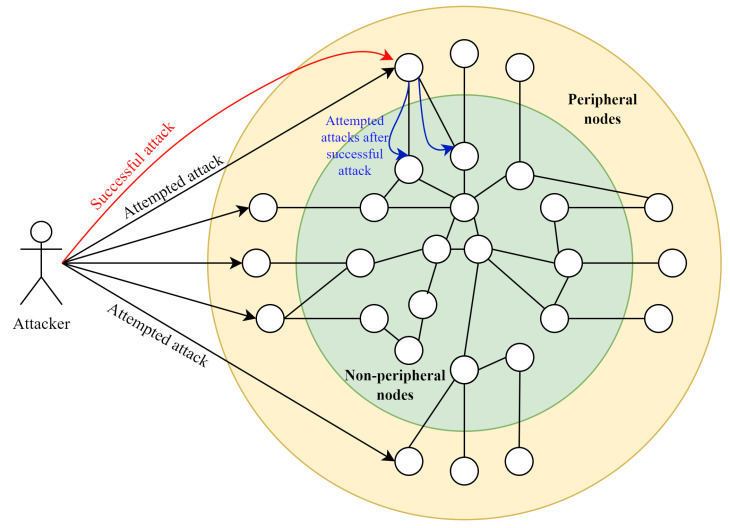
Modeling the network and attackers. The nodes in the yellow region are peripheral, and thus attackers can launch attempted attacks, shown as black arrows. The nodes in the green region are not directly accessible to the attackers and can only be accessed, shown as blue arrows, after successful attack on a peripheral node, shown as a red arrow.

**Figure 4 entropy-26-00384-f004:**
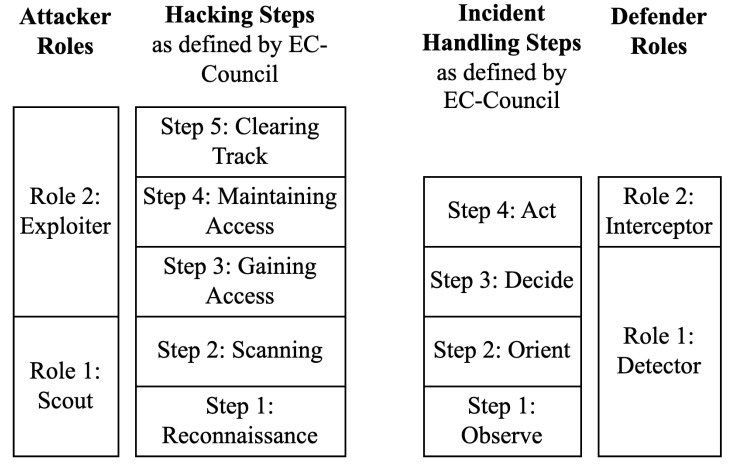
Attacker and defender roles matched to EC-Council’s hacking and incident handling steps. Hacking (penetration) follows a series of 5 steps, which are reconnaissance, scanning, gaining access, maintaining access and clearing tracks. We define two attacker roles; the first is scout, which performs the first two hacking steps before reporting the findings to the second role, exploiter, which performs the last three hacking steps. Incident handling follows a series of 4 steps, which are observe, orient, decide and act. We define two defender roles; the first is detector, which performs the first three incident hacking steps before reporting the findings to the second role, interceptor, which performs the last incident handling steps.

**Figure 5 entropy-26-00384-f005:**
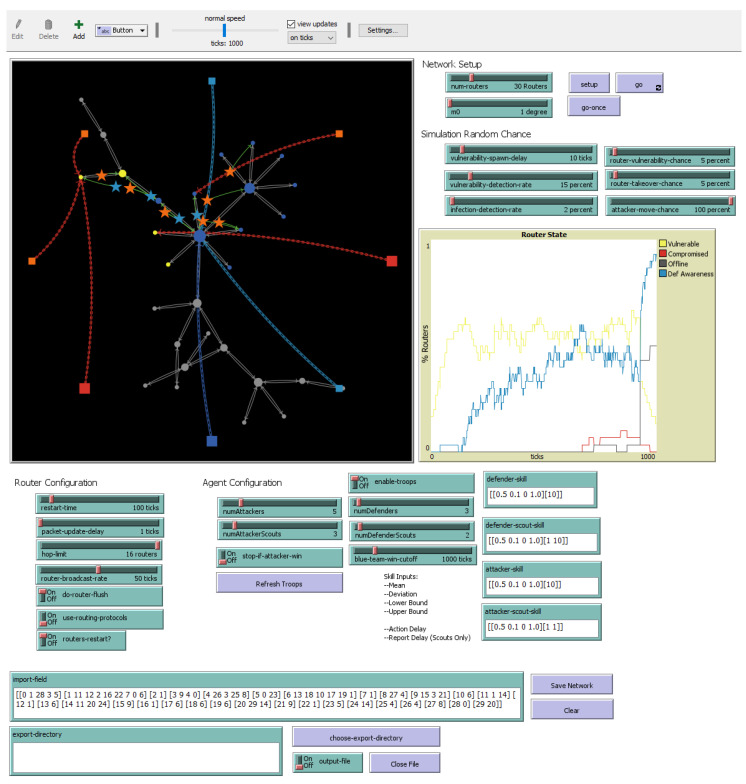
CCTFv2 simulation control panel. The panel can be divided into inputs (sliders and buttons) and outputs (graph, plots and output monitors). The inputs resemble the control variables described in Figure 2 and detailed in Table 1 and Table 2. The outputs resemble the measured variables.

**Figure 6 entropy-26-00384-f006:**
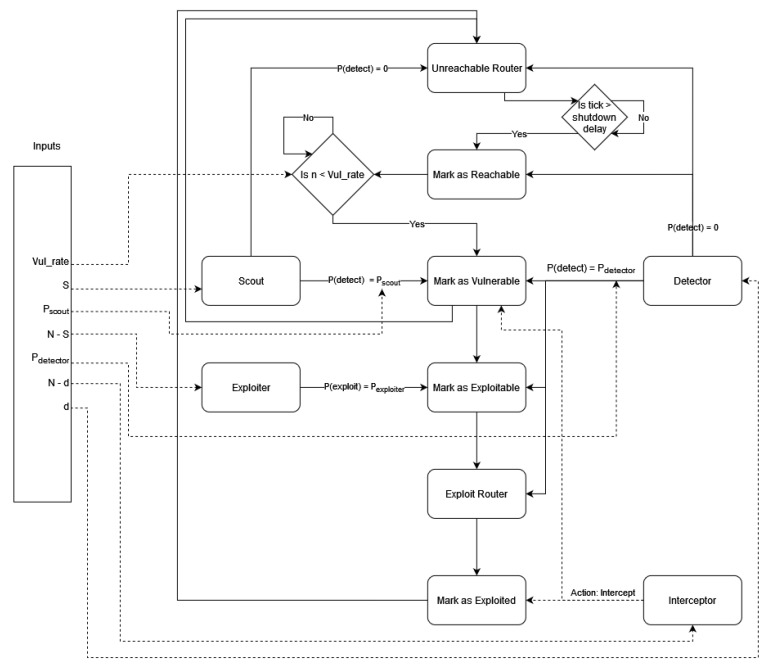
Simulation agent logic flowchart. The left rectangles represent the attacker roles, while the right rectangles represent the defender roles. The attackers and defenders alter the statuses of the network nodes, which are described in the central rectangles.

**Figure 7 entropy-26-00384-f007:**
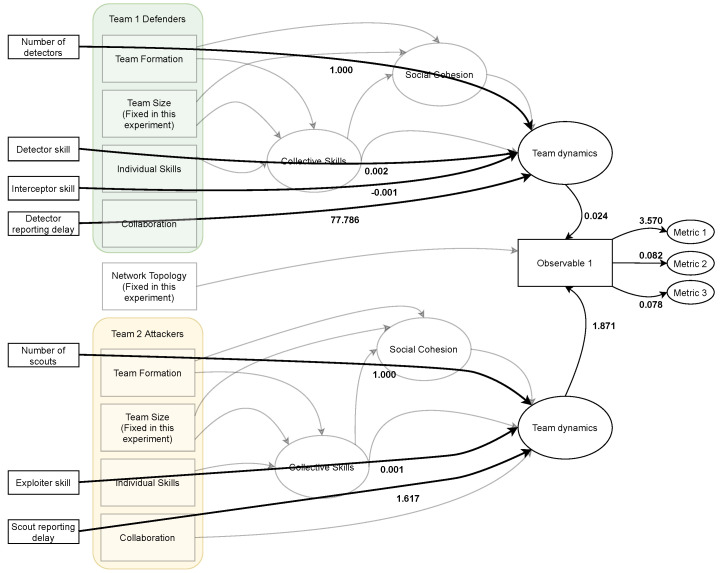
Theoretical model verification (multi-layer structural equation model). The bold lines describe the relationship between the controlled and latent variables and the latent and observed variables.

**Figure 8 entropy-26-00384-f008:**
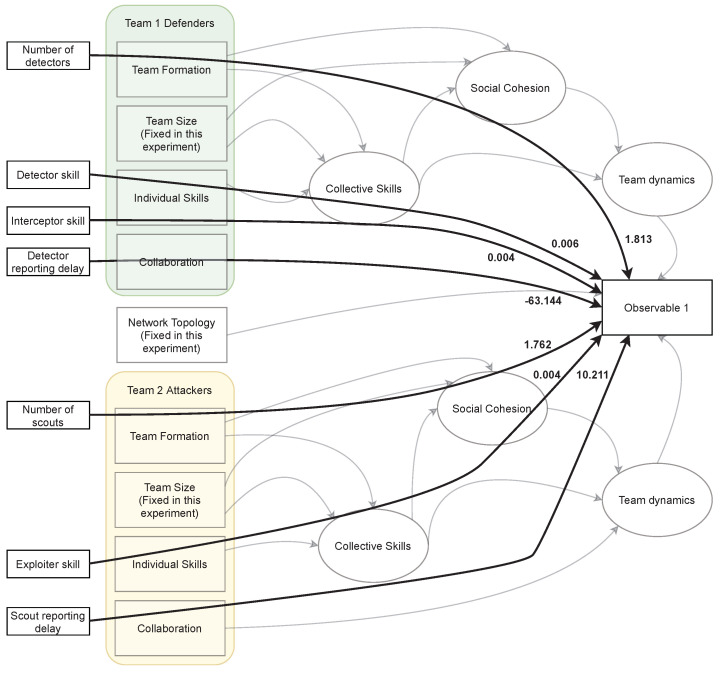
Two-layer structural equation model. The bold lines describe the relationship between the controlled and observed variables.

**Table 1 entropy-26-00384-t001:** Training input parameters.

Input Parameter	Individual Simulation Values	Description	Baseline
*R*	Not changed	Number of routers in network.	100
*N*	Not changed	Total number of team members (*N*) for each side.	10
*S*	1–9	Number of attacker scouts (*S*). 1≤S<N. Exploiters: N−S.	5
*d*	1-9	Number of defender detectors (*d*). 1≤d<N. Interceptors: N−d.	2
Vul_round	1–100	Simulation ticks between vulnerabilities being generated.	1
Vul_rate	Not changed	Chance of a router becoming vulnerable when vulnerabilities are generated.	5%
$P_{scout}$	Not changed	Probability of attacker scout discovering a vulnerable router.	100%
$P_{exploiter}$	1–100%	Probability of exploiter to exploit an exploitable router.	5
$P_{detector-vulnerable}$	1-100	Probability of defender detector discovering a vulnerable router.	15
$P_{detector-exploited}$	1–100	Probability of defender detector discovering an exploited router.	20
$\Delta_{interceptor}$	1–200 ticks	Time needed to restore a router.	100 ticks
Agent_Skill	N(0.25,0.1),N(0.5,0.1),N(0.75,0.1)	Multiplier for successful action probability.	*N* (0.5, 0.1)
Action_Delay	1,5,10,25 ticks	Ticks between agent actions.	1 tick
Report_Delay	1,5,10,25 ticks	Ticks between detector and scout findings and reports.	1 tick

**Table 2 entropy-26-00384-t002:** Validation/testing input parameters.

Input Parameter	Description	Simulation Values
*R*	Number of routers in network.	40, 60
*N*	Total number of team members (*N*) for each side.	5
*S*	Number of attacker scouts (*S*). 1≤S<N. Exploiters: N−S.	2, 3
*d*	Number of defender detectors (*d*). 1≤d<N. Interceptors: N−d.	2, 3
Vul_round	Simulation ticks between vulnerabilities being generated.	1
Vul_rate	Chance of a router becoming vulnerable when vulnerabilities are generated.	1, 3, 5, 7, 10%
$P_{scout}$	Probability of attacker scout discovering a vulnerable router.	100%
$P_{exploiter}$	Probability of exploiter to exploit an exploitable router.	5%
$P_{detector-vulnerable}$	Probability of defender detector discovering a vulnerable router.	15%
$P_{detector-exploited}$	Probability of defender detector discovering an exploited router.	1, 3, 5%
$\Delta_{interceptor}$	Time needed to restore a router.	100 ticks
Agent_Skill	Multiplier for successful action probability.	N(0.5,0.1)
Action_Delay	Ticks between agent actions.	1 tick
Report_Delay	Ticks between detector and scout findings and reports.	1,5,10 ticks

## Data Availability

CCTF-framework: Collaborative Cyber Team Formation Framework—Netlogo Simulation. Available online: https://github.com/Starwhip/CCTF-Framework (accessed on 24 April 2024).

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
