# Peer review of "CCTFv2: Modeling Cyber Competitions"

_entropy, 2024, doi:10.3390/e26050384_

Round 1

Reviewer 1 Report

Comments and Suggestions for Authors

The paper considers the problem of team performance and team formation in the context of cybersecurity and cyber competitions. It presents an agent-based approach to team formation and its influence on the team performance which is an important task. The paper has a good structure, clear text and acceptable graphic material.

The following fixes and additions are recommended:

1. In abstract you should show your main results with ther novelty. 

2. Novelty of the aproach is not clearly expressed in the introduction.

3. In Section 2 (Related Work) it is recommended to add more relevant works that use agent-based teamwork models. For example you could add to Related work the following papers:

Gaston M.E.4 DesJardins M. The effect of network structure on dynamic team formation in multi‐agent systems. Computational Intelligence, 2008.

Dignum F.; Dunin-Keplicz B.; Verbrugge R. Agent theory for team formation by dialogue. Intelligent Agents VII, Springer. 2001.

Marcolino LS; Jiang AX; Tambe M. Multi-agent team formation: Diversity beats strength? IJCAI, 2013 - projects.iq.harvard.edu

Kotenko, I. Multi-agent simulation of attacks and defense mechanisms in computer networks. International Journal of Computing. 2008, Vol. 7, Issue 2, 35-43. DOI: 10.47839/ijc.7.2.508.

4. The description of experiments is shallow. Each feagure should be clearly explained. 

5. There is no comparison of the experimental results with other works.

6. Conlusion should be extended by clear specification of the results of experiments and future research of the authors. 

Slips:

p.2: %80 -> 80%

p.3: In [32], the authors -> In [32], the author

Comments on the Quality of English Language

Minor editing of English language required.

Author Response

Attached please find the responses to the referees.

Reviewer 2 Report

Comments and Suggestions for Authors

The abstract should present something about the simulations made in NetLogo and its results.

Speaking about the 2002 French national team should start a new paragraph (see line 45).

Authors must fill in lines [485] – [498] with their information or delete these lines if they are not applicable to their article.

Reference [15] does not include the publication year.

Authors should give more details regarding references [22] - [25], [73] and [77]: the information source, authors, appearance year etc.

Author Response

(The authors gave the same response as above.)

Reviewer 3 Report

Comments and Suggestions for Authors

In this paper, the authors present a study on cybersecurity teams' performance using Agent-Based Modeling. 

The paper covers an interesting topic, it follows a previous work by the same authors, and its English is generally well-written. However, the authors are encouraged to check the following:

- The abstract should include 1-context, 2-problem/opportunity, 3-proposal/contribution, and 4-main results and conclusions. Thus, the abstract should be improved regarding (3) and (4), i.e. to clearly identify the contribution and to present the main results regarding team performance, before the conclusions.

- Authors in the abstract refer that: "Cybersecurity activities (...) are intrinsically collective and rely on teams as ("an") indispensable factor". Authors should better contextualize this sentence. This generic assumption may not hold in specific contexts.

- In the Introduction section instead of "%80" it should be "80%"

- In the Introduction section instead of "offenders" it should be "forwards"

- The paper should not have forward references in sections such as "As discussed in section 2, cyber team formation did not receive sufficient..." 

- In the introduction authors present the CCTFv2 model, however, nothing is said regarding the CCTFv1. The reference to this previous work appears first and shortly in section 3 as reference 50. Thus, it is unclear what is the relation and the differences between v1 and v2.

- The main idea of the model is not clearly explained in the introduction section. In the introduction section, authors should explain generally what is being proposed and save details for the following sections, in a more structured and clear explanation.

- Authors should avoid subjective expressions such as "(...) our research contributes significantly (...)"

- Authors should check if they intended to to write "some" in the following expression: "section 2 covers some of the important works"

- The related work section should have an introduction for this section, not regarding the contribution. In this section, the contributions from other authors are analyzed and detailed. The gaps should be identified at the end of each or multiple contributions from other authors.

- To describe the contribution in [37] the name OSIRIS is not relevant

- The "network attack modelling" is not introduced in the first part of section 2.

- Fig 6 could be presented in vector format

- The paper does compare the results of this model with other models

- The paper misses the analysis of the results presented in section 4

- The paper misses a discussion section on the results and analysis retrieved from this study

- The conclusion could be improved to be less short and vague

Comments on the Quality of English Language

Included in "Comments for authors"

Author Response

(The authors gave the same response as above.)

Round 2

Reviewer 1 Report

Comments and Suggestions for Authors

The authors responded to the comments. The quality of figures is low (esp. figures 5 and 6).